# Learning Neural Generative Dynamics for Molecular Conformation Generation

**Minkai Xu\***[1,2]**, Shitong Luo\***[3]**, Yoshua Bengio**[1,2,4]**, Jian Peng**[5]**, Jian Tang**[1,4,6]

[1]Mila - Québec AI Institute, Canada
[2]Université de Montréal, Canada
[3]Peking University, China
[4]Canadian Institute for Advanced Research (CIFAR), Canada
[5]University of Illinois at Urbana-Champaign, USA
[6]HEC Montréal, Canada
{xuminkai,yoshua.bengio}@mila.quebec
luost@pku.edu.cn
jianpeng@illinois.edu
jian.tang@hec.ca

## Abstract

We study how to generate molecule conformations (*i.e.*, 3D structures) from a molecular graph. Traditional methods, such as molecular dynamics, sample conformations via computationally expensive simulations. Recently, machine learning methods have shown great potential by training on a large collection of conformation data. Challenges arise from the limited model capacity for capturing complex distributions of conformations and the difficulty in modeling long-range dependencies between atoms. Inspired by the recent progress in deep generative models, in this paper, we propose a novel probabilistic framework to generate valid and diverse conformations given a molecular graph. We propose a method combining the advantages of both flow-based and energy-based models, enjoying: (1) a high model capacity to estimate the multimodal conformation distribution; (2) explicitly capturing the complex long-range dependencies between atoms in the observation space. Extensive experiments demonstrate the superior performance of the proposed method on several benchmarks, including conformation generation and distance modeling tasks, with a significant improvement over existing generative models for molecular conformation sampling[1].

## 1 Introduction

Recently, we have witnessed the success of graph-based representations for molecular modeling in a variety of tasks such as property prediction (Gilmer et al., 2017) and molecule generation (You et al., 2018; Shi et al., 2020). However, a more natural and intrinsic representation of a molecule is its 3D structure, commonly known as the molecular geometry or *conformation*, which represents each atom by its 3D coordinate. The conformation of a molecule determines its biological and physical properties such as charge distribution, steric constraints, as well as interactions with other molecules. Furthermore, large molecules tend to comprise a number of rotatable bonds, which may induce flexible conformation changes and a large number of feasible conformations in nature. Generating valid and stable conformations of a given molecule remains very challenging. Experimentally, such structures are determined by expensive and time-consuming crystallography. Computational approaches based on Markov chain Monte Carlo (MCMC) or molecular dynamics (MD) (De Vivo et al., 2016) are computationally expensive, especially for large molecules (Ballard et al., 2015).

Machine learning methods have recently shown great potential for molecular conformation generation by training on a large collection of data to model the probability distribution of potential conformations $\boldsymbol{R}$ based on the molecular graph $\mathcal{G}$, *i.e.*, $p(\boldsymbol{R}|\mathcal{G})$. For example, Mansimov et al.

---

*Equal contribution. Work was done during Shitong's internship at Mila.

[1]Code is available at https://github.com/DeepGraphLearning/CGCF-ConfGen.

(2019) proposed a Conditional Variational Graph Autoencoders (CVGAE) for molecular conformation generation. A graph neural network (Gilmer et al., 2017) is first applied to the molecular graph to get the atom representations, based on which 3D coordinates are further generated. One limitation of such an approach is that by directly generating the 3D coordinates of atoms it fails to model the rotational and translational invariance of molecular conformations. To address this issue, instead of generating the 3D coordinates directly, Simm & Hernández-Lobato (2020) recently proposed to first model the molecule's distance geometry (*i.e.*, the distances between atoms)—which are rotationally and translationally invariant—and then generate the molecular conformation based on the distance geometry through a post-processing algorithm (Liberti et al., 2014). Similar to Mansimov et al. (2019), a few layers of graph neural networks are applied to the molecular graph to learn the representations of different *edges*, which are further used to generate the distances of different edges independently. This approach is capable of more often generating valid molecular conformations.

Although these new approaches have made tremendous progress, the problem remains very challenging and far from solved. First, each molecule may have multiple stable conformations around a number of states which are thermodynamically stable. In other words, the distribution $p(\boldsymbol{R}|\mathcal{G})$ is very complex and multi-modal. Models with high capacity are required to model such complex distributions. Second, existing approaches usually apply a few layers of graph neural networks to learn the representations of nodes (or edges) and then generate the 3D coordinates (or distances) based on their representations independently. Such approaches are necessarily limited to capturing a single mode of $p(\boldsymbol{R}|\mathcal{G})$ (since the coordinates or distances are sampled independently) and are incapable of modeling multimodal joint distributions and the form of the graph neural net computation makes it difficult to capture long-range dependencies between atoms, especially in large molecules.

Inspired by the recent progress with deep generative models, this paper proposes a novel and principled probabilistic framework for molecular geometry generation, which addresses the above two limitations. Our framework combines the advantages of normalizing flows (Dinh et al., 2014) and energy-based approaches (LeCun et al., 2006), which have a strong model capacity for modeling complex distributions, are flexible to model long-range dependency between atoms, and enjoy efficient sampling and training procedures. Similar to the work of Simm & Hernández-Lobato (2020), we also first learn the distribution of distances $\boldsymbol{d}$ given the graph $\mathcal{G}$, *i.e.*, $p(\boldsymbol{d}|\mathcal{G})$, and define another distribution of conformations $\boldsymbol{R}$ given the distances $\boldsymbol{d}$, *i.e.*, $p(\boldsymbol{R}|\boldsymbol{d}, \mathcal{G})$. Specifically, we propose a novel Conditional Graph Continuous Flow (CGCF) for distance geometry ($\boldsymbol{d}$) generation conditioned on the molecular graph $\mathcal{G}$. Given a molecular graph $\mathcal{G}$, CGCF defines an invertible mapping between a base distribution (*e.g.*, a multivariate normal distribution) and the molecular distance geometry, using a virtually infinite number of graph transformation layers on atoms represented by a Neural Ordinary Differential Equations architecture (Chen et al., 2018). Such an approach enjoys very high flexibility to model complex distributions of distance geometry. Once the molecular distance geometry $\boldsymbol{d}$ is generated, we further generate the 3D coordinates $\boldsymbol{R}$ by searching from the probability $p(\boldsymbol{R}|\boldsymbol{d}, \mathcal{G})$.

Though the CGCF has a high capacity for modeling complex distributions, the distances of different edges are still independently updated in the transformations, which limits its capacity for modeling long-range dependency between atoms in the sampling process. Therefore, we further propose another unnormalized probability function, *i.e.*, an energy-based model (EBM) (Hinton & Salakhutdinov, 2006; LeCun et al., 2006; Ngiam et al., 2011), which acts as a tilting term of the flow-based distribution and directly models the joint distribution of $\boldsymbol{R}$. Specifically, the EBM trains an energy function $E(\boldsymbol{R}, \mathcal{G})$, which is approximated by a neural network. The flow- and energy-based models are combined in a novel way for joint training and mutual enhancement. First, energy-based methods are usually difficult to train due to the slow sampling process. In addition, the distribution of conformations is usually highly multi-modal, and the sampling procedures based on Gibbs sampling or Langevin Dynamics (Bengio et al., 2013a;b) tend to get trapped around modes, making it difficult to mix between different modes (Bengio et al., 2013a). Here we use the flow-based model as a proposal distribution for the energy model, which is capable to generate diverse samples for training energy models. Second, the flow-based model lacks the capacity to explicitly model the long-range dependencies between atoms, which we find can however be effectively modeled by an energy function $E(\boldsymbol{R}, \mathcal{G})$. Our sampling process can be therefore viewed as a *two-stage dynamic* system, where we first take the flow-based model to quickly synthesize realistic conformations and then used the learned energy $E(\boldsymbol{R}, \mathcal{G})$ to refine the generated conformations through Langevin Dynamics.

We conduct comprehensive experiments on several recently proposed benchmarks, including GEOM-QM9, GEOM-Drugs (Axelrod & Gomez-Bombarelli, 2020) and ISO17 (Simm & Hernández-Lobato, 2020). Numerical evaluations show that our proposed framework consistently outperforms the previous state-of-the-art (GraphDG) on both conformation generation and distance modeling tasks, with a clear margin.

## 2 PROBLEM DEFINITION AND PRELIMINARIES

### 2.1 PROBLEM DEFINITION

**Notations.** Following existing work (Simm & Hernández-Lobato, 2020), each molecule is represented as an undirected graph $\mathcal{G} = \langle \mathcal{V}, \mathcal{E} \rangle$, where $\mathcal{V}$ is the set of nodes representing atoms and $\mathcal{E}$ is the set of edges representing inter-atomic bonds. Each node $v$ in $\mathcal{V}$ is labeled with atomic properties such as element type. The edge in $\mathcal{E}$ connecting $u$ and $v$ is denoted as $e_{uv}$, and is labeled with its bond type. We also follow the previous work (Simm & Hernández-Lobato, 2020) to expand the molecular graph with auxiliary bonds, which is elaborated in Appendix B. For the molecular 3D representation, each atom in $\mathcal{V}$ is assigned with a 3D position vector $\boldsymbol{r} \in \mathbb{R}^3$. We denote $d_{uv} = \|\boldsymbol{r}_u - \boldsymbol{r}_v\|_2$ as the Euclidean distance between the $u^{th}$ and $v^{th}$ atom. Therefore, we can represent all the positions $\{\boldsymbol{r}_v\}_{v \in \mathcal{V}}$ as a matrix $\boldsymbol{R} \in \mathbb{R}^{|\mathcal{V}| \times 3}$ and all the distances between connected nodes $\{d_{uv}\}_{e_{uv} \in \mathcal{E}}$ as a vector $\boldsymbol{d} \in \mathbb{R}^{|\mathcal{E}|}$.

**Problem Definition.** The problem of *molecular conformation generation* is defined as a conditional generation process. More specifically, our goal is to model the conditional distribution of atomic positions $\boldsymbol{R}$ given the molecular graph $\mathcal{G}$, *i.e.*, $p(\boldsymbol{R}|\mathcal{G})$.

### 2.2 PRELIMINARIES

**Continuous Normalizing Flow.** A normalizing flow (Dinh et al., 2014; Rezende & Mohamed, 2015) defines a series of invertible deterministic transformations from an initial known distribution $p(z)$ to a more complicated one $p(x)$. Recently, normalizing flows have been generalized from discrete number of layers to continuous (Chen et al., 2018; Grathwohl et al., 2018) by defining the transformation $f_\theta$ as a continuous-time dynamic $\frac{\partial z(t)}{\partial t} = f_\theta(z(t), t)$. Formally, with the latent variable $z(t_0) \sim p(z)$ at the start time, the continuous normalizing flow (CNF) defines the transformation $x = z(t_0) + \int_{t_0}^{t_1} f_\theta(z(t), t) dt$. Then the exact density for $p_\theta(x)$ can be computed by:

$$\log p_\theta(x) = \log p(z(t_0)) - \int_{t_0}^{t_1} \text{Tr}\left(\frac{\partial f_\theta}{\partial z(t)}\right) dt \qquad (1)$$

where $z(t_0)$ can be obtained by inverting the continuous dynamic $z(t_0) = x + \int_{t_1}^{t_0} f_\theta(z(t), t) dt$. A black-box ordinary differential equation (ODE) solver can be applied to estimate the outputs and inputs gradients and optimize the CNF model (Chen et al., 2018; Grathwohl et al., 2018).

**Energy-based Models.** Energy-based models (EBMs) (Dayan et al., 1995; Hinton & Salakhutdinov, 2006; LeCun et al., 2006) use a scalar parametric energy function $E_\phi(x)$ to fit the data distribution. Formally, the energy function induces a density function with the Boltzmann distribution $p_\phi(x) = \exp(-E_\phi(x))/Z(\phi)$, where $Z = \int \exp(-E_\phi(x)) dx$ denotes the partition function. EBM can be learned with Noise contrastive estimation (NCE) (Gutmann & Hyvärinen, 2010) by treating the normalizing constant as a free parameter. Given the training examples from both the dataset and a noise distribution $q(x)$, $\phi$ can be estimated by maximizing the following objective function:

$$J(\phi) = \mathbb{E}_{p_{\text{data}}}\left[\log \frac{p_\phi(x)}{p_\phi(x) + q(x)}\right] + \mathbb{E}_q\left[\log \frac{q(x)}{p_\phi(x) + q(x)}\right], \qquad (2)$$

which turns the estimation of EBM into a discriminative learning problem. Sampling from $E_\phi$ can be done with a variety of methods such as Markov chain Monte Carlo (MCMC) or Gibbs sampling (Hinton & Salakhutdinov, 2006), possibly accelerated using Langevin dynamics (Du & Mordatch, 2019; Song et al., 2020), which leverages the gradient of the EBM to conduct sampling:

$$x_k = x_{k-1} - \frac{\epsilon}{2}\nabla_x E_\phi(x_{k-1}) + \sqrt{\epsilon}\omega, \omega \sim \mathcal{N}(0, \mathcal{I}), \qquad (3)$$

where $\epsilon$ refers to the step size. $x_0$ are the samples drawn from a random initial distribution and we take the $x_K$ with $K$ Langevin dynamics steps as the generated samples of the stationary distribution.

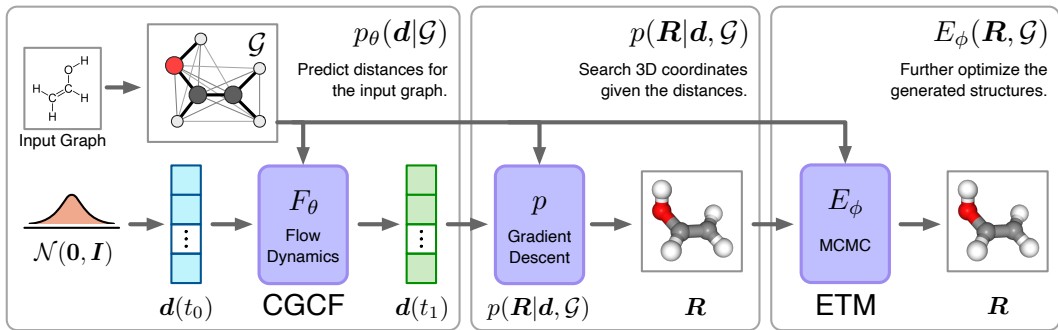

Figure 1: Illustration of the proposed framework. Given the molecular graph, we 1) first draw latent variables from a Gaussian prior, and transform them to the desired distance matrix through the Conditional Graph Continuous Flow (CGCF); 2) search the possible 3D coordinates according to the generated distances and 3) further optimize the generated conformation via a MCMC procedure with the Energy-based Tilting Model (ETM).

## 3 METHOD

### 3.1 OVERVIEW

We first present a high-level description of our model. Directly learning a generative model on Cartesian coordinates heavily depends on the (arbitrary) rotation and translation (Mansimov et al., 2019). Therefore, in this paper we take the atomic pairwise distances as intermediate variables to generate conformations, which are invariant to rotation and translation. More precisely, the cornerstone of our method is to factorize the conditional distribution $p_\theta(\boldsymbol{R}|\mathcal{G})$ into the following formulation:

$$p_\theta(\boldsymbol{R}|\mathcal{G}) = \int p(\boldsymbol{R}|\boldsymbol{d}, \mathcal{G}) \cdot p_\theta(\boldsymbol{d}|\mathcal{G}) \, \mathrm{d}\boldsymbol{d}, \tag{4}$$

where $p_\theta(\boldsymbol{d}|\mathcal{G})$ models the distribution of inter-atomic distances given the graph $\mathcal{G}$ and $p(\boldsymbol{R}|\boldsymbol{d}, \mathcal{G})$ models the distribution of conformations given the distances $\boldsymbol{d}$. In particular, the conditional generative model $p_\theta(\boldsymbol{d}|\mathcal{G})$ is parameterized as a conditional graph continuous flow, which can be seen as a continuous dynamics system to transform the random initial noise to meaningful distances. This flow model enables us to capture the long-range dependencies between atoms in the hidden space during the dynamic steps.

Though CGCF can capture the dependency between atoms in the hidden space, the distances of different edges are still independently updated in the transformations, which limits the capacity of modeling the dependency between atoms in the sampling process. Therefore we further propose to correct $p_\theta(\boldsymbol{R}|\mathcal{G})$ with an energy-based tilting term $E_\phi(\boldsymbol{R}, \mathcal{G})$:

$$p_{\theta,\phi}(\boldsymbol{R}|\mathcal{G}) \propto p_\theta(\boldsymbol{R}|\mathcal{G}) \cdot \exp(-E_\phi(\boldsymbol{R}, \mathcal{G})). \tag{5}$$

The tilting term is directly defined on the joint distribution of $\boldsymbol{R}$ and $\mathcal{G}$, which explicitly captures the long-range interaction directly in observation space. The tilted distribution $p_{\theta,\phi}(\boldsymbol{R}|\mathcal{G})$ can be used to provide refinement or optimization for the conformations generated from $p_\theta(\boldsymbol{R}|\mathcal{G})$. This energy function is also designed to be invariant to rotation and translation.

In the following parts, we will firstly describe our flow-based generative model $p_\theta(\boldsymbol{R}|\mathcal{G})$ in Section 3.2 and elaborate the energy-based tilting model $E_\phi(\boldsymbol{R}, \mathcal{G})$ in Section 3.3. Then we introduce the two-stage sampling process with both deterministic and stochastic dynamics in Section 3.4. An illustration of the whole framework is given in Fig. 1.

### 3.2 FLOW-BASED GENERATIVE MODEL

**Conditional Graph Continuous Flows** $p_\theta(\boldsymbol{d}|\mathcal{G})$. We parameterize the conditional distribution of distances $p_\theta(\boldsymbol{d}|\mathcal{G})$ with the continuous normalizing flow, named **C**onditional **G**raph **C**ontinuous **F**low (CGCF). CGCF defines the distribution through the following dynamics system:

$$\boldsymbol{d} = F_\theta(\boldsymbol{d}(t_0), \mathcal{G}) = \boldsymbol{d}(t_0) + \int_{t_0}^{t_1} f_\theta(\boldsymbol{d}(t), t; \mathcal{G})\mathrm{d}t, \quad \boldsymbol{d}(t_0) \sim \mathcal{N}(\boldsymbol{0}, \boldsymbol{I}) \tag{6}$$

where the dynamic $f_\theta$ is implemented by Message Passing Neural Networks (MPNN) (Gilmer et al., 2017), which is a widely used architecture for representation learning on molecular graphs. MPNN takes node attributes, edge attributes and the bonds lengths $\boldsymbol{d}(t)$ as input to compute the node and edge embeddings. Each message passing layer updates the node embeddings by aggregating the information from neighboring nodes according to its hidden vectors of respective nodes and edges. Final features are fed into a neural network to compute the value of the dynamic $f_\theta$ for all distances independently. As $t_1 \to \infty$, our dynamic can have an infinite number of steps and is capable to model long-range dependencies. The invertibility of $F_\theta$ allows us to not only conduct fast sampling, but also easily optimize the parameter set $\theta$ by minimizing the exact negative log-likelihood:

$$\mathcal{L}_{\mathrm{mle}}(\boldsymbol{d}, \mathcal{G}; \theta) = -\mathbb{E}_{p_{\mathrm{data}}} \log p_\theta(\boldsymbol{d}|\mathcal{G}) = -\mathbb{E}_{p_{\mathrm{data}}} \left[ \log p(\boldsymbol{d}(t_0)) + \int_{t_0}^{t_1} \mathrm{Tr} \left( \frac{\partial f_{\theta, G}}{\partial \boldsymbol{d}(t)} \right) dt \right]. \quad (7)$$

**Closed-form $p(\boldsymbol{R}|\boldsymbol{d}, \mathcal{G})$.** The generated pair-wise distances can be converted into 3D structures through postprocessing methods such as the classic Euclidean Distance Geometry (EDG) algorithm. In this paper, we adopt an alternative way by defining the conformations as a conditional distribution:

$$p(\boldsymbol{R}|\boldsymbol{d}, \mathcal{G}) = \frac{1}{Z} \exp \left\{ - \sum_{e_{uv} \in \mathcal{E}} \alpha_{uv} \left( \|\boldsymbol{r}_u - \boldsymbol{r}_v\|_2 - d_{uv} \right)^2 \right\}, \quad (8)$$

where $Z$ is the partition function to normalize the probability and $\{\alpha_{uv}\}$ are parameters that control the variance of desired Cartesian coordinates, which can be either learned or manually designed according to the graph structure $\mathcal{G}$. With the probabilistic formulation, we can conduct either sampling via MCMC or searching the local optimum with optimization methods. This simple function is fast to calculate, making the generation procedure very efficient with a negligible computational cost.

Compared with the conventional EDG algorithm adopted in GraphDG (Simm & Hernández-Lobato, 2020), our probabilistic solution enjoys following advantages: 1) $p(\boldsymbol{R}|\boldsymbol{d}, \mathcal{G})$ enables the calculation for the likelihood $p_\theta(\boldsymbol{R}|\mathcal{G})$ of Eq. 4 by approximation methods, and thus can be further combined with the energy-based tilting term $E_\phi(\boldsymbol{R}, \mathcal{G})$ to induce a superior distribution; 2) GraphDG suffers the drawback that when invalid sets of distances are generated, EDG will fail to construct 3D structure. By contrast, our method can always be successful to generate conformations by sampling from the distribution $p(\boldsymbol{R}|\boldsymbol{d}, \mathcal{G})$.

### 3.3 ENERGY-BASED TILTING MODEL

The last part of our framework is the **E**nergy-based **T**ilting **M**odel (ETM) $E_\phi(\boldsymbol{R}, \mathcal{G})$, which helps model the long-range interactions between atoms explicitly in the observation space. $E_\phi(\boldsymbol{R}, \mathcal{G})$ takes the form of SchNet (Schütt et al., 2017), which is widely used to model the potential-energy surfaces and energy-conserving force fields for molecules. The continuous-filter convolutional layers in SchNet allow each atom to aggregate the representations of all single, pairwise, and higher-order interactions between the atoms through non-linear functions. The final atomic representations are pooled to a single vector and then passed into a network to produce the scalar output.

Typically the EBMs can be learned by maximum likelihood, which usually requires the lengthy MCMC procedure and is time-consuming for training. In this work, we learn the ETM by Noise Contrastive Estimation (Gutmann & Hyvärinen, 2010), which is much more efficient. In practice, the noise distribution is required to be close to data distribution, otherwise the classification problem would be too easy and would not guide $E_\phi$ to learn much about the modality of the data. We propose to take the pre-trained CGCF to serve as a strong noise distribution, leading to the following discriminative learning objective for the ETM[2]:

$$\mathcal{L}_{\mathrm{nce}}(\boldsymbol{R}, \mathcal{G}; \phi) = - \mathbb{E}_{p_{\mathrm{data}}} \left[ \log \frac{1}{1 + \exp(E_\phi(\boldsymbol{R}, \mathcal{G}))} \right] - \mathbb{E}_{p_\theta} \left[ \log \frac{1}{1 + \exp(-E_\phi(\boldsymbol{R}, \mathcal{G}))} \right]. \quad (9)$$

### 3.4 SAMPLING

We employ a two-stage dynamic system to synthesize a possible conformation given the molecular graph representation $\mathcal{G}$. In the first stage, we first draw a latent variable $\hat{z}$ from the Gaussian prior

---

[2]Detailed derivations of the training loss can be found in Appendix F.

$\mathcal{N}(0, I)$, and then pass it through the continuous deterministic dynamics model $F_\theta$ defined in Eq. 6 to get $\hat{d}_0 = F_\theta(\hat{z}_0, G)$. Then an optimization procedure such as stochastic gradient descent is employed to search the realistic conformations $\boldsymbol{R}$ with local maximum probability of $p(\boldsymbol{R}|\boldsymbol{d}, \mathcal{G})$ (defined in Eq. 8). By doing this, an initial conformation $\boldsymbol{R}^{(0)}$ can be generated. In the second stage, we further refine the initial conformation $\boldsymbol{R}^{(0)}$ with the energy-based model defined in Eq. 5 with $K$ steps of Langevin dynamics:

$$
\begin{aligned}
&\boldsymbol{R}_k = \boldsymbol{R}_{k-1} - \frac{\epsilon}{2} \nabla_{\boldsymbol{R}} E_{\theta,\phi}(\boldsymbol{R}|\mathcal{G}) + \sqrt{\epsilon}\omega, \omega \sim \mathcal{N}(0, \mathcal{I}), \\
&\text{where } E_{\theta,\phi}(\boldsymbol{R}|\mathcal{G}) = -\log p_{\theta,\phi}(\boldsymbol{R}|\mathcal{G}) = E_\phi(\boldsymbol{R}, \mathcal{G}) - \log \int p(\boldsymbol{R}|\boldsymbol{d}, \mathcal{G}) p_\theta(\boldsymbol{d}|\mathcal{G}) \mathrm{d}\boldsymbol{d}.
\end{aligned}
\tag{10}
$$

where $\epsilon$ denotes the step size. The second integration term in $E_{\theta,\phi}$ can be estimated through approximate methods. In practice, we use Monte Carlo Integration to conduct the approximation, which is simple yet effective with just a few distance samples from the CGCF model $p_\theta(\boldsymbol{d}|\mathcal{G})$.

## 4    EXPERIMENTS

### 4.1    EXPERIMENT SETUP

**Evaluation Tasks.** To evaluate the performance of proposed model, we conduct experiments by comparing with the counterparts on: (1) **Conformation Generation** evaluates the model's capacity to learn the distribution of conformations by measuring the diversity and accuracy of generated samples (section 4.2); (2) **Distribution over distances** is first proposed in Simm & Hernández-Lobato (2020), which concentrate on the distance geometry of generated conformations (section 4.2).

**Benchmarks.** We use the recent proposed GEOM-QM9 and GEOM-Drugs (Axelrod & Gomez-Bombarelli, 2020) datasets for conformation generation task and ISO17 dataset (Simm & Hernández-Lobato, 2020) for distances modeling task. The choice of different datasets is because of their distinct properties. Specifically, GEOM datasets consist of stable conformations, which is suitable to evaluate the conformation generation task. By contrast, ISO17 contains snapshots of molecular dynamics simulations, where the structures are not equilibrium conformations but can reflect the density around the equilibrium state. Therefore, it is more suitable for the assessment of similarity between the model distribution and the data distribution around equilibrium states.

More specifically, **GEOM-QM9** is an extension to the QM9 (Ramakrishnan et al., 2014) dataset: it contains multiple conformations for most molecules while the original QM9 only contains one. This dataset is limited to 9 heavy atoms (29 total atoms), with small molecular mass and few rotatable bonds. We randomly draw 50000 conformation-molecule pairs from GEOM-QM9 to be the training set, and take another 17813 conformations covering 150 molecular graphs as the test set. **GEOM-Drugs** dataset consists of much larger drug molecules, up to a maximum of 181 atoms (91 heavy atoms). It also contains multiple conformations for each molecule, with a larger variance in structures, *e.g.*, there are the 6.5 rotatable bonds in average. We randomly take 50000 conformation-molecule pairs from GEOM-Drugs as the training set, and another 9161 conformations (covering 100 molecular graphs) as the test split. **ISO17** dataset is also built upon QM9 datasets, which consists of 197 molecules, each with 5000 conformations. Following Simm & Hernández-Lobato (2020), we also split ISO17 into the training set with 167 molecules and the test set with another 30 molecules.

**Baselines**. We compared our proposed method with the following state-of-the-art conformation generation methods. **CVGAE** (Mansimov et al., 2019) uses a conditional version of VAE to directly generate the 3D coordinates of atoms given the molecular graph. **GraphDG** (Simm & Hernández-Lobato, 2020) also employs the conditional VAE framework. Instead of directly modeling the 3D structure, they propose to learn the distribution over distances. Then the distances are converted into conformations with an EDG algorithm. Furthermore, we also take **RDKit** (Riniker & Landrum, 2015) as a baseline model, which is a classical EDG approach built upon extensive calculation collections in computational chemistry.

Table 1: Comparison of different methods on the COV and MAT scores. Top 4 rows: deep generative models for molecular conformation generation. Bottom 5 rows: different methods that involve an additional rule-based force field to further optimize the generated structures.

| Dataset | GEOM-QM9 | | | | GEOM-Drugs | | | |
|---|---|---|---|---|---|---|---|---|
| Metric | COV* (%) | | MAT (Å) | | COV* (%) | | MAT (Å) | |
| | Mean | Median | Mean | Median | Mean | Median | Mean | Median |
| CVGAE | 8.52 | 5.62 | 0.7810 | 0.7811 | 0.00 | 0.00 | 2.5225 | 2.4680 |
| GraphDG | 55.09 | 56.47 | 0.4649 | 0.4298 | 7.76 | 0.00 | 1.9840 | 2.0108 |
| **CGCF** | 69.60 | 70.64 | 0.3915 | 0.3986 | 49.92 | 41.07 | 1.2698 | 1.3064 |
| **CGCF + ETM** | **72.43** | **74.38** | **0.3807** | **0.3955** | **53.29** | **47.06** | **1.2392** | **1.2480** |
| RDKit | **79.94** | **87.20** | 0.3238 | **0.3195** | 65.43 | 70.00 | 1.0962 | 1.0877 |
| CVGAE + FF | 63.10 | 60.95 | 0.3939 | 0.4297 | 83.08 | 95.21 | 0.9829 | 0.9177 |
| GraphDG + FF | 70.67 | 70.82 | 0.4168 | 0.3609 | 84.68 | 93.94 | 0.9129 | 0.9090 |
| **CGCF + FF** | **73.52** | **72.75** | 0.3131 | 0.3251 | 92.28 | 98.15 | **0.7740** | **0.7338** |
| **CGCF + ETM + FF** | **73.54** | 72.58 | **0.3088** | **0.3210** | **92.41** | **98.57** | 0.7737 | 0.7616 |

\* For the reported COV score, the threshold $\delta$ is set as 0.5Å for QM9 and 1.25Å for Drugs. More results of COV scores with different threshold $\delta$ are given in Appendix H.

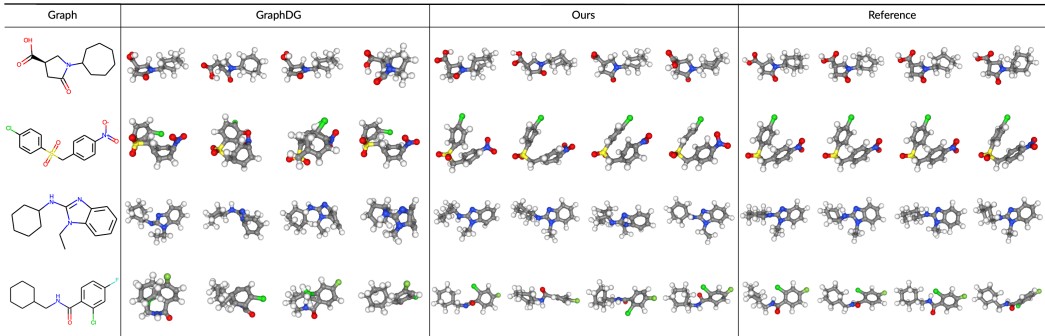

Figure 2: Visualization of generated conformations from the state-of-the-art baseline (GraphDG), our method and the ground-truth, based on four random molecular graphs from the test set of GEOM-Drugs. C, O, H, S and Cl are colored gray, red, white, yellow and green respectively.

## 4.2 CONFORMATION GENERATION

In this section, we evaluate the ability of the proposed method to model the equilibrium conformations. We focus on both the *diversity* and *accuracy* of the generated samples. More specifically, diversity measures the model's capacity to generate multi-modal conformations, which is essential for discovering new conformations, while accuracy concentrates on the similarity between generated conformations and the equilibrium conformations.

**Evaluation.** For numerical evaluations, we follow previous work (Hawkins, 2017; Mansimov et al., 2019) to calculate the Root-Mean-Square Deviation (RMSD) of the heavy atoms between generated samples and reference ones. Precisely, given the generated conformation $\boldsymbol{R}$ and the reference $\boldsymbol{R}^*$, we obtain $\hat{\boldsymbol{R}}$ by translating and rotating $\boldsymbol{R}^*$ to minimize the following predefined RMSD metric:

$$\text{RMSD}(\boldsymbol{R}, \hat{\boldsymbol{R}}) = \left(\frac{1}{n}\sum_{i=1}^{n}\|\boldsymbol{R}_i - \hat{\boldsymbol{R}}_i\|^2\right)^{\frac{1}{2}}, \tag{11}$$

where $n$ is the number of heavy atoms. Then the smallest distance is taken as the evaluation metric. Built upon the RMSD metric, we define **Cov**erage (COV) and **Mat**ching (MAT) score to measure the diversity and quality respectively. Intuitively, COV measures the fraction of conformations in the reference set that are matched by at least one conformation in the generated set. For each conformation in the generated set, its neighbors in the reference set within a given RMSD threshold

Table 2: Comparison of distances density modeling with different methods. We compare the marginal distribution of single ($p(d_{uv}|\mathcal{G})$), pair ($p(d_{uv}, d_{ij}|\mathcal{G})$) and all ($p(\boldsymbol{d}|\mathcal{G})$) edges between C and O atoms. Molecular graphs $\mathcal{G}$ are taken from the test set of ISO17. We take two metrics into consideration: 1) **median** MMD between the ground truth and generated ones, and 2) **mean** ranking (1 to 3) based on the MMD metric.

|  | Single | | Pair | | All | |
|---|---|---|---|---|---|---|
|  | Mean | Median | Mean | Median | Mean | Median |
| RDKit | 3.4513 | 3.1602 | 3.8452 | 3.6287 | 4.0866 | 3.7519 |
| CVGAE | 4.1789 | 4.1762 | 4.9184 | 5.1856 | 5.9747 | 5.9928 |
| GraphDG | 0.7645 | 0.2346 | 0.8920 | 0.3287 | 1.1949 | 0.5485 |
| **CGCF** | **0.4490** | **0.1786** | **0.5509** | **0.2734** | **0.8703** | **0.4447** |
| **CGCF + ETM** | 0.5703 | 0.2411 | 0.6901 | 0.3482 | 1.0706 | 0.5411 |

$\delta$ are marked as matched:

$$\text{COV}(\mathbb{S}_g(\mathcal{G}), \mathbb{S}_r(\mathcal{G})) = \frac{1}{|\mathbb{S}_r|} \Big| \Big\{ \boldsymbol{R} \in \mathbb{S}_r \, \big| \, \text{RMSD}(\boldsymbol{R}, \boldsymbol{R}') < \delta, \exists \boldsymbol{R}' \in \mathbb{S}_g \Big\} \Big|, \qquad (12)$$

where $\mathbb{S}_g(\mathcal{G})$ denotes the generated conformations set for molecular graph $\mathcal{G}$, and $\mathbb{S}_r(\mathcal{G})$ denotes the reference set. In practice, the number of samples in the generated set is two times of the reference set. Typically, a higher COV score means the a better diversity performance. The COV score is able to evaluate whether the generated conformations are diverse enough to cover the ground-truth.

While COV is effective to measure the diversity and detect the mode-collapse case, it is still possible for the model to achieve high COV with a high threshold tolerance. Here we define the MAT score as a complement to measure the quality of generated samples. For each conformation in the reference set, the RMSD distance to its nearest neighbor in the generated set is computed and averaged:

$$\text{MAT}(\mathbb{S}_g(\mathcal{G}), \mathbb{S}_r(\mathcal{G})) = \frac{1}{|\mathbb{S}_r|} \sum_{\boldsymbol{R}' \in \mathbb{S}_r} \min_{\boldsymbol{R} \in \mathbb{S}_g} \text{RMSD}(\boldsymbol{R}, \boldsymbol{R}'). \qquad (13)$$

This metric concentrate on the accuracy of generated conformations. More realistic generated samples lead to a lower matching score.

**Results.** Tab. 1 shows that compared with the existing state-of-the-art baselines, our CGCF model can already achieve superior performance on all four metrics (top 4 rows). As a CNF-based model, CGCF holds much the higher generative capacity for both diversity and quality compared than VAE approaches. The results are further improved when combined with ETM to explicitly incorporate the long-range correlations. We visualize several representative examples in Fig. 2, and leave more examples in Appendix G. A meaningful observation is that though competitive over other neural models, the rule-based RDKit method occasionally shows better performance than our model, which indicates that RDKit can generate more realistic structures. We argue that this is because after generating the initial coordinates, RDKit involves additional hand-designed molecular force field (FF) energy functions (Rappé et al., 1992; Halgren, 1996a) to find the stable conformations with local minimal energy. By contrast, instead of finding the local minimums, our deep generative models aim to model and sample from the potential distribution of structures. To yield a better comparison, we further test our model by taking the generated structures as initial states and utilize the Merck Molecular Force Field (MMFF) (Halgren, 1996a) to find the local stable points. A more precise description of about the MMFF Force Field algorithms in RDKit is given in Appendix I. This postprocessing procedure is also employed in the previous work (Mansimov et al., 2019). Additional results in Tab. 1 verify our conjecture that FF plays a vital role in generating more realistic structures, and demonstrate the capacity of our method to generate high-quality initial coordinates.

### 4.3 DISTRIBUTIONS OVER DISTANCES

Tough primarily designed for 3D coordinates, we also following Simm & Hernández-Lobato (2020) to evaluate the generated distributions of pairwise distance, which can be viewed as a representative element of the model capacity to model the inter-atomic interactions.

**Evaluation.** Let $p(d_{uv}|\mathcal{G})$ denote the conditional distribution of distances on each edge $e_{uv}$ given a molecular graph $\mathcal{G}$. The set of distances are computed from the generated conformations $\boldsymbol{R}$. We

Table 3: Conformation Diversity. Mean and Std represent the corresponding mean and standard deviation of pairwise RMSD between the generated conformations per molecule.

|      | RDKit | CVGAE | GraphDG | CGCF | CGCF +ETM |
|------|-------|-------|---------|------|-----------|
| Mean | 0.083 | 0.207 | 0.249   | 0.810 | 0.741     |
| Std  | 0.054 | 0.187 | 0.104   | 0.223 | 0.206     |

calculate maximum mean discrepancy (MMD) (Gretton et al., 2012) to compare the generated distributions and the ground-truth distributions. Specifically, we evaluate the distribution of individual distances $p(d_{uv}|\mathcal{G})$, pair distances $p(d_{uv}, d_{ij}|\mathcal{G})$ and all distances $p(\boldsymbol{d}|\mathcal{G})$. For this benchmark, the number of samples in the generated set is the same as the reference set.

**Results.** The results of MMD are summarized in Tab. 2. The statistics show that RDKit suffers the worst performance, which is because it just aims to generate the most stable structures as illustrated in Section 4.2. For CGCF, the generated samples are significantly closer to the ground-truth distribution than baseline methods, where we consistently achieve the best numerical results. Besides, we notice that ETM will slightly hurt the performance in this task. However, one should note that this phenomenon is natural because typically ETM will sharpen the generated distribution towards the stable conformations with local minimal energy. By contrast, the ISO17 dataset consists of snapshots of molecular dynamics where the structures are not equilibrium conformations but samples from the density around the equilibrium state. Therefore, ETM will slightly hurt the results. This phenomenon is also consistent with the observations for RDKit. Instead of generating unbiased samples from the underlying distribution, RDKit will only generate the stable ones with local minimal energy by involving the hand-designed molecular force field (Simm & Hernández-Lobato, 2020). And as shown in the results, though highly competitive in Tab. 1, RDKit also suffers much weaker results in Tab. 2. The marginal distributions $P(d_{uv}|\mathcal{G})$ for pairwise distances in visualized in Appendix K, which further demonstrate the superior capacity of our proposed method.

We also follow Mansimov et al. (2019) to calculate the diversity of conformations generated by all compared methods, which is measured by calculating the mean and standard deviation of the pairwise RMSD between each pair of generated conformations per molecule. The results shown in Tab. 3 demonstrate that while our method can achieve the lowest MMD, it does not collapse to generating extremely similar conformations. Besides, we observe that ETM will slightly hurt the diversity of CGCF, which verifies our statement that ETM will sharpen the generated distribution towards the stable conformations with local minimal energy.

## 5 CONCLUSION AND FUTURE WORK

In this paper, we propose a novel probabilistic framework for molecular conformation generation. Our generative model combines the advantage of both flow-based and energy-based models, which is capable of modeling the complex multi-modal geometric distribution and highly branched atomic correlations. Experimental results show that our method outperforms all previous state-of-the-art baselines on the standard benchmarks. Future work includes applying our framework on much larger datasets and extending it to more challenging structures (*e.g.*, proteins).

## ACKNOWLEDGMENTS

This project is supported by the Natural Sciences and Engineering Research Council (NSERC) Discovery Grant, the Canada CIFAR AI Chair Program, collaboration grants between Microsoft Research and Mila, Samsung Electronics Co., Ldt., Amazon Faculty Research Award, Tencent AI Lab Rhino-Bird Gift Fund and a NRC Collaborative R&D Project (AI4D-CORE-06). This project was also partially funded by IVADO Fundamental Research Project grant PRF-2019-3583139727.

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

# A    RELATED WORKS

**Conformation Generation.** There have been results showing deep learning speeding up molecular dynamics simulation by learning efficient alternatives to quantum mechanics-based energy calculations (Schütt et al., 2017; Smith et al., 2017). However, though accelerated by neural networks, these approaches are still time-consuming due to the lengthy MCMC process. Recently, Gebauer et al. (2019) and Hoffmann & Noé (2019) propose to directly generate 3D structures with deep generative models. However, these models can hardly capture graph- or bond-based structure, which is typically complex and highly branched. Some other works (Lemke & Peter, 2019; AlQuraishi, 2019; Ingraham et al., 2019; Noé et al., 2019; Senior et al., 2020) also focus on learning models to directly generate 3D structure, but focus on the protein folding problem. Unfortunately, proteins are linear structures while general molecules are highly branched, making these methods not naturally transferable to general molecular conformation generation tasks.

**Energy-based Generative Model.** There has been a long history for energy-based generative models. Xie et al. (2016) proposes to train an energy-based model parameterized by modern deep neural network and learned it by Langevin based MLE. The model is called generative ConvNet since it can be derived from the discriminative ConvNet. In particular, this paper is the first to formulate modern ConvNet-parametrized EBM as exponential tilting of a reference distribution, and connect it to discriminative ConvNet classifier. More recently, Du & Mordatch (2019) implemented the deep EBMs with ConvNet as energy function and achieved impressive results on image generation.

Different from the previous works, we concentrate on molecular geometry generation, and propose a novel and principled probabilistic framework to address the domain-specific problems. More specifically, we first predict the atomic distances through the continuous normalizing flow, and then convert them to the desired 3D conformation and optimize it with the energy-based model. This procedure enables us to keep the rotational and translational invariance property. Besides, to the best of our knowledge, we are the first one to combine neural ODE with EBMs. We take the ODE model to improve the training of EBM, and combine both to conduct the two-stage sampling dynamics.

# B    DATA PREPROCESS

Inspired by classic molecular distance geometry (Crippen et al., 1988), we also generate the confirmations by firstly predicting all the pairwise distances, which enjoys the invariant property to rotation and translation. Since the bonds existing in the molecular graph are not sufficient to characterize a conformation, we pre-process the graphs by extending *auxiliary* edges. Specifically, the atoms that are 2 or 3 hops away are connected with *virtual bonds*, labeled differently from the real bonds of the original graph. These extra edges contribute to reducing the degrees of freedom in the 3D coordinates, with the edges between second neighbors helping to fix the angles between atoms, and those between third neighbors fixing dihedral angles.

# C    NETWORK ARCHITECTURE

In this section, we elaborate on the network architecture details of CGCF and ETM.

## C.1    CONTINUOUS GRAPH FLOW

In CGCF, the dynamic function $f_\theta$ defined in Eq. 6 is instanced with a message passing neural networks. Given the node attributes, edge attributes and intermediate edge lengths as input, we first embed them into the feature space through feedforward networks:

$$
\begin{aligned}
\boldsymbol{h}_v^{(0)} &= \mathrm{NodeEmbedding}(v), \quad v \in \mathcal{V}, \\
\boldsymbol{h}_{e_{uv}} &= \mathrm{EdgeEmbedding}(e_{uv}, d_{uv}(t_0)), \quad e_{uv} \in \mathcal{E}.
\end{aligned}
\tag{14}
$$

Then, the node and edge features along are passed sequentially into $L$ layers message passing networks with the graph structure $\mathcal{G}$:

$$
\boldsymbol{h}_v^{(\ell)} = \mathrm{MLP}\left(\boldsymbol{h}_v^{(\ell-1)} + \sum_{u \in N_{\mathcal{G}}(v)} \sigma(\boldsymbol{h}_u^{(\ell-1)} + \boldsymbol{h}_{e_{uv}})\right), \quad \ell = 1 \ldots L,
\tag{15}
$$

where $N_{\mathcal{G}}(v)$ denotes the first neighbors in the graph $\mathcal{G}$ and $\sigma$ is the activation function. After $L$ message passing layers, we use the final hidden representation $h^{(L)}$ as the node representations. Then for each bond, the corresponding node features are aggregated along with the edge feature to be fed into a neural network to compute the value of the dynamic $f_\theta$:

$$\frac{\partial d_{uv}}{\partial t} = \text{NN}(\boldsymbol{h}_u, \boldsymbol{h}_v, \boldsymbol{h}_{e_{uv}}, t). \tag{16}$$

### C.2 ENERGY-BASED TILTING MODEL

The ETM is implemented with SchNet. It takes both the graph and conformation information as input and output a scalar to indicate the energy level. Let the atoms are described by a tuple of features $\mathbf{X}^l = (\mathbf{x}_1^l, \ldots, \mathbf{x}_n^l)$, where $n$ denote the number of atoms and $l$ denote the layer. Then given the positions $\boldsymbol{R}$, the node embeddings are updated by the convolution with all surrounding atoms:

$$\mathbf{x}_i^{l+1} = \left(X^l * W^l\right)_i = \sum_{j=0}^{n_{\text{atoms}}} \mathbf{x}_j^l \circ W^l \left(\mathbf{r}_j - \mathbf{r}_i\right), \tag{17}$$

where "o" represents the element-wise multiplication. It is straightforward that the above function enables to include translational and rotational invariance by computing pairwise distances instead of using relative positions. After $L$ convolutional layers, we perform a sum-pooling operator over the node embeddings to calculate the global embedding for the whole molecular structure. Then the global embedding is fed into a feedforward network to compute the scalar of the energy level.

## D  TWO-STAGE DYNAMIC SYSTEM FOR SAMPLING

---
**Algorithm 1** Sampling Procedure of the Proposed Method

---
**Input**: molecular graph $\mathcal{G}$, CGCF model with parameter $\theta$, ETM with parameter $\phi$, the number of optimization steps for $p(\boldsymbol{R}|\boldsymbol{d}, \mathcal{G})$ $M$ and its step size $r$, the number of MCMC steps for $E_{\theta,\phi}$ $N$ and its step size $\epsilon$
**Output**: molecular conformation $\boldsymbol{R}$
1: Sample $\boldsymbol{d}(t_0) \sim \mathcal{N}(0, \mathcal{I})$
2: $\boldsymbol{d} = F_\theta(\boldsymbol{d}(t_0), \mathcal{G})$
3: **for** $m = 1, ..., M$ **do**
4:     $\boldsymbol{R}_m = \boldsymbol{R}_{m-1} + r\nabla_{\boldsymbol{R}} \log p(\boldsymbol{R}|\boldsymbol{d}, \mathcal{G})$
5: **end for**
6: **for** $n = 1, ..., N$ **do**
7:     $\boldsymbol{R}_n = \boldsymbol{R}_{n-1} - \frac{\epsilon}{2}\nabla_{\boldsymbol{R}} E_{\theta,\phi}(\boldsymbol{R}|\mathcal{G}) + \sqrt{\epsilon}\omega, \omega \sim \mathcal{N}(0, \mathcal{I}),$
8: **end for**

---

## E  IMPLEMENTATION DETAILS

Our model is implemented in PyTorch (Paszke et al., 2017). The MPNN in CGCF is implemented with 3 layers, and the embedding dimension is set as 128. And the SchNet in ETM is implemented with 6 layers with the embedding dimension set as 128. We train our CGCF with a batch size of 128 and a learning rate of 0.001 until convergence. After obtaining the CGCF, we train the ETM with a batch size of 384 and a learning rate of 0.001 until convergence. For all experimental settings, we use Adam (Kingma & Ba, 2014) to optimize our model.

## F   DETAILED DERIVATIONS OF ENERGY-BASED MODEL

Here we present the detailed derivations of the training objective function of Energy-based Tilting Model (ETM) in Eq. 9:

$$
\begin{aligned}
\mathcal{L}_{\text{nce}}(\boldsymbol{R}, \mathcal{G}; \phi) = & -\mathbb{E}_{p_{\text{data}}}\Big[\log \frac{p_{\theta,\phi}(\boldsymbol{R}|\mathcal{G})}{p_{\theta,\phi}(\boldsymbol{R}|\mathcal{G}) + p_{\theta}(\boldsymbol{R}|\mathcal{G})}\Big] - \mathbb{E}_{p_{\theta}}\Big[\log \frac{p_{\theta}(\boldsymbol{R}|\mathcal{G})}{p_{\theta,\phi}(\boldsymbol{R}|\mathcal{G}) + p_{\theta}(\boldsymbol{R}|\mathcal{G})}\Big] \\
= & -\mathbb{E}_{p_{\text{data}}}\Big[\log \frac{p_{\theta}(\boldsymbol{R}|\mathcal{G})\exp(-E_{\phi}(\boldsymbol{R}, \mathcal{G}))}{p_{\theta}(\boldsymbol{R}|\mathcal{G})\exp(-E_{\phi}(\boldsymbol{R}, \mathcal{G})) + p_{\theta}(\boldsymbol{R}|\mathcal{G})}\Big] \\
& -\mathbb{E}_{p_{\theta}}\Big[\log \frac{p_{\theta}(\boldsymbol{R}|\mathcal{G})}{p_{\theta}(\boldsymbol{R}|\mathcal{G})\exp(-E_{\phi}(\boldsymbol{R}, \mathcal{G})) + p_{\theta}(\boldsymbol{R}|\mathcal{G})}\Big] \\
= & -\mathbb{E}_{p_{\text{data}}}\Big[\log \frac{1}{1 + \exp(E_{\phi}(\boldsymbol{R}, \mathcal{G}))}\Big] - \mathbb{E}_{p_{\theta}}\Big[\log \frac{1}{1 + \exp(-E_{\phi}(\boldsymbol{R}, \mathcal{G}))}\Big].
\end{aligned}
\tag{18}
$$

## G   MORE GENERATED SAMPLES

We present more visualizations of generated 3D structures in Fig. 3, which are generated from our model (CGCF + ETM) learned on both GEOM-QM9 and GEOM-Drugs datasets. The visualizations demonstrate that our proposed framework holds the high capacity to model the chemical structures in the 3D coordinates.

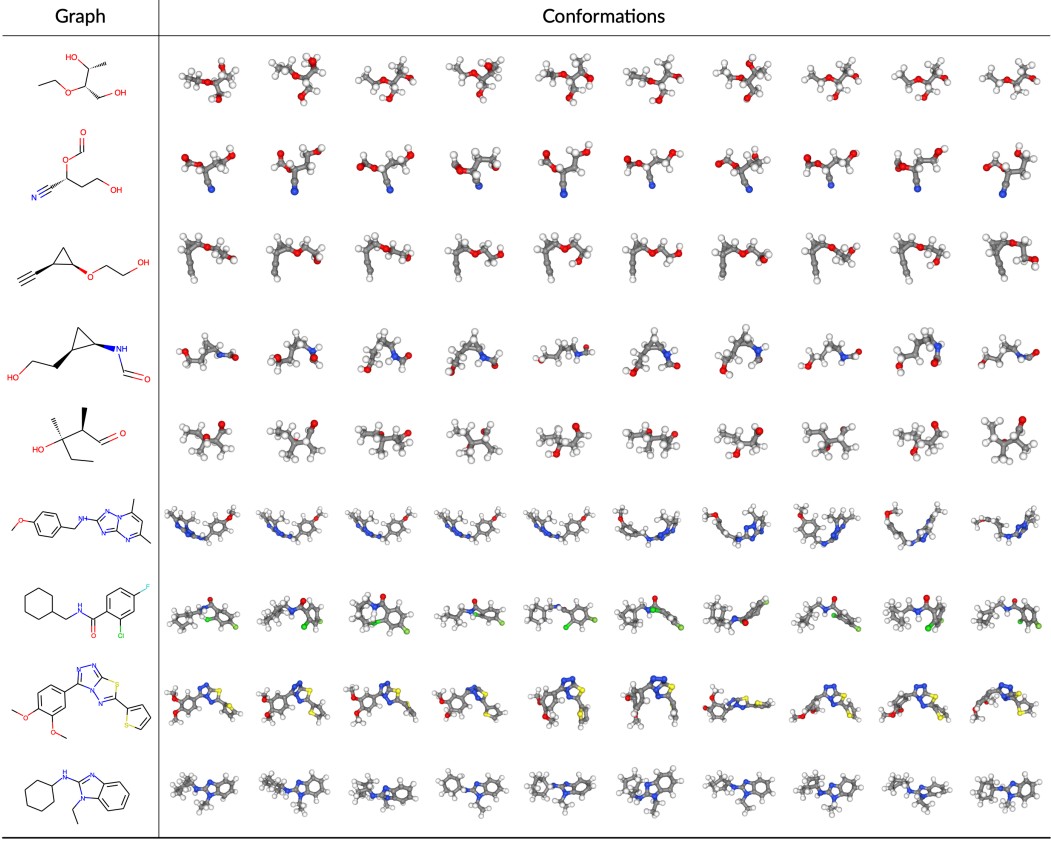

Figure 3: Visualizations of generated graphs from our proposed method. In each row, we show multiple generated conformations for one molecular graph. For the top 5 rows, the graphs are chosen from the small molecules in GEOM-QM9 test dataset; and for the bottom 4 rows, graphs are chosen from the larger molecules in GEOM-Drugs test dataset. C, O, H, S and CI are colored gray, red, white, yellow and green respectively.

## H    MORE RESULTS OF COVERAGE SCORE

We give more results of the coverage (COV) score with different threshold $\delta$ in Fig. 4. As shown in the figure, our proposed method can consistently outperform the previous state-of-the-art baselines CVGAE and GraphDG, which demonstrate the effectiveness of our model.

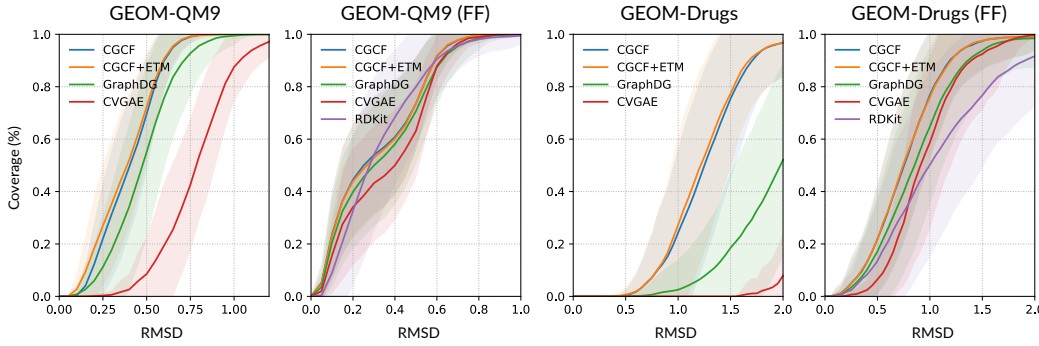

Figure 4: Curves of the averaged coverage score with different RMSD thresholds on GEOM-QM9 (left two) and GEOM-Drugs (right two) datasets. The first and third curves are results of only the generative models, while the other two are results when further optimized with rule-based force fields.

## I    IMPLEMENTATION FOR MMFF

In this section, we give a more precise description of the MMFF Force Field implementation in the RDKit toolkit (Riniker & Landrum, 2015).

In MMFF, the energy expression is constituted by seven terms: bond stretching, angle bending, stretch-bend, out-of-plane bending, torsional, van der Waals and electrostatic. The detailed functional form of individual terms can be found in the original literature (Halgren, 1996a). To build the force field for a given molecular system, the first step is to assign correct types to each atom. At the second step, atom-centered partial charges are computed according to the MMFF charge model (Halgren, 1996b). Then, all bonded and non-bonded interactions in the molecular system under study, depending on its structure and connectivity, are loaded into the energy expression. Optionally, external restraining terms can be added to the MMFF energy expression, with the purpose of constraining selected internal coordinates during geometry optimizations. Once all bonded and non-bonded interactions, plus optional restraints, have been loaded into the MMFF energy expression, potential gradients of the system under study can be computed to minimize the energy.

## J    MORE EVALUATIONS FOR CONFORMATION GENERATION

**Junk Rate.** The COV and MAT score in Section 4.2 do not appear to explicitly measure the generated false samples. Here we additionally define **Junk** rate measurement. Intuitively, JUNK measures the fraction of generated conformations that are far away from all the conformations in the reference set. For each conformation in the generated set, it will be marked as a false sample if its RMSD to all the conformations of reference set are above a given threshold $\delta$:

$$\text{JUNK}(\mathbb{S}_g(\mathcal{G}), \mathbb{S}_r(\mathcal{G})) = \frac{1}{|\mathbb{S}_g|}\left|\left\{\boldsymbol{R} \in \mathbb{S}_g \,\big|\, \text{RMSD}(\boldsymbol{R}, \boldsymbol{R}') > \delta, \forall \boldsymbol{R}' \in \mathbb{S}_r\right\}\right|, \qquad (19)$$

Typically, a lower JUNK rate means better generated quality. The results are shown in Tab. 4. As shown in the table, our CGCF model can already outperform the existing state-of-the-art baselines with an obvious margin. The results are further improved when combined with ETM to explicitly incorporate the long-range correlations.

## K    DISTANCE DISTRIBUTION VISUALIZATION

In Fig. 5, we plot the marginal distributions $p(d_{uv}|\mathcal{G})$ for all pairwise distances between C and O atoms of a molecular graph in the ISO17 test set. As shown in the figure, though primarily designed

Table 4: Comparison of different methods on the JUNK scores. Top 4 rows: deep generative models for molecular conformation generation. Bottom 5 rows: different methods that involve an additional rule-based force field to further optimize the generated structures.

| Dataset | GEOM-QM9 | | GEOM-Drugs | |
| --- | --- | --- | --- | --- |
| Metric | JUNK* (%) | | JUNK* (%) | |
| | Mean | Median | Mean | Median |
| CVGAE | 71.59 | 100.00 | 100.00 | 100.00 |
| GraphDG | 61.25 | 66.26 | 97.83 | 100.00 |
| **CGCF** | 55.24 | 57.24 | 77.82 | 90.00 |
| **CGCF + ETM** | **52.15** | **54.23** | **75.81** | **88.64** |
| RDKit | 17.07 | 5.90 | 45.51 | 45.94 |
| CVGAE + FF | 62.92 | 71.21 | 72.01 | 78.44 |
| GraphDG + FF | 45.53 | 46.35 | 55.50 | 61.54 |
| **CGCF + FF** | 43.01 | 46.69 | 37.48 | 36.63 |
| **CGCF + ETM + FF** | **41.63** | **43.97** | **36.16** | **33.05** |

* For the reported JUNK score, the threshold $\delta$ is set as 0.5Å for QM9 and 1.25Å for Drugs.

for 3D structure generation, our method can make much better estimation of the distances than GraphDG, which is the state-of-the-art model for molecular geometry prediction. As a representative element of the pairwise property between atoms, the inter-atomic distances demonstrate the capacity of our model to capture the inter-atomic interactions.

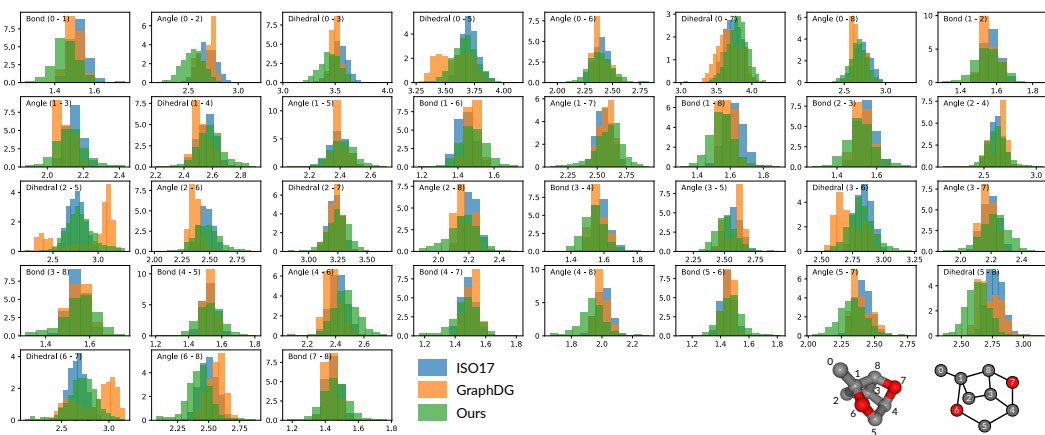

Figure 5: Marginal distributions $p(d_{uv}|\mathcal{G})$ of ground-truth and generated conformations between C and O atoms given a molecular graph from the test set of ISO17. In each subplot, the annotation $(u - v)$ indicates the atoms connected by the corresponding bond $d_{uv}$. We concentrate on the heavy atoms (C and O) and omit the H atoms for clarity.

