# OpenReview forum: "Learning Neural Generative Dynamics for Molecular Conformation Generation"
_ICLR.cc/2021/Conference — ICLR 2021 Poster_

### Official Review · AnonReviewer1 · 2020-10-28
**Creative modeling**

**Rating:** 6
**Confidence:** 3

**Review:**

This paper combines flow-based and energy-based models to generate molecular conformations from a molecular graph.

Strengths
- The modeling is well-motivated and creative. A continuous flow model maps a graph-based molecular representation into a distribution over conformations. The continuous flow is well-motivated; it factors out the distance calculations to resolve concerns in previous works where pairwise distances are modeled independently.
- Additionally, an energy-based model (EBM) is used to further help the model capture long-range atomic interactions. Recent advancements in EBMs are deployed for stable training, such as a lavengin dynamics.
- The paper is well-written, clear and easy to follow, despite the complexity of the model
- Strong baselines: CVGAE, GraphDG, and RDKit. Results are slightly better than CVGAE for conformation generation, and slightly better for distance distribution.

Weaknesses
- While the model performs better than neural baselines, it does not yet approach RDKit, which is based on expert knowledge features. This means the proposed model is not yet practically useful.
- Only two tasks were investigated, and only one or two datasets within each task. The empirical evaluation of this paper could be improved by including more experiments (e.g. along the lines of Mansimov et al. 2019)
- The paper attributes the advantage in RDKit to its inclusion of a force field. Therefore, the authors combine a force field with their model and assess the difference (good!) However, the conclusion in the paper seems misguided - looking at Table 1, including the force field does not result in a significant change in their model's performance. In fact, this experiment was also done in the CVGAE paper (Mansimov et al. 2019).
- In Table 1, the EBM improves performance on the conformation generation tasks. However, in Table 2, the EBM decreases performance on the distribution over distances task. I would have liked to see more investigation or discussion of why this is the case.
- The RDKit baseline is not included in Table 2. It is included in the GraphDG paper and should be included here as well.

Overall, this paper presents a creative model that performs better than existing neural models. However, the evaluation feels preliminary compared to what we see in earlier works.

---

> ### Author Response · Authors · 2020-11-17
> **Response to the AnonReviewer1**
>
> Thanks for your constructive comments and suggestions. The response to your concerns are listed below:
>
> Q1: While the model performs better than neural baselines, it does not yet approach RDKit, which is based on expert knowledge features. This means the proposed model is not yet practically useful.
> A1: 1. As also claimed in Simm et al., methods such as RDKit just aim to produce a low-energy conformation, not to generate unbiased samples from the underlying distribution at a certain temperature. By contrast, our deep generative models can model the underlying distribution and generate samples from it. This advantage can be observed from distribution Tab 2, where both GraphDG and our method can outperform RDKit with a significant margin.
> 2. Besides, on the large molecule dataset Drugs, when combined with the force field (FF), our method achieved very good results (COV 92.41 and MAT 0.77), which is much better than RDKit.  Moreover, the Drugs dataset is also more challenging than QM9, which indicates that the proposed method is already practically useful for the large and complex conformations.
>
> Q2: Only two tasks were investigated, and only one or two datasets within each task. The empirical evaluation of this paper could be improved by including more experiments (e.g. along the lines of Mansimov et al. 2019)
> A2: Thanks for your suggestions and we have added three experiments:
> 1). In Section 4.3, we follow Mansimov et al. to add a new experiment to measure the diversity of generated conformations, which is a statistic over the generated distribution instead of over the test set. The results (on ISO17) are measured by calculating the mean and standard deviation of the pairwise RMSD between each pair of generated conformations per molecule.
> 2). In Appendix J, we follow Axelrod et al. to add an experiment on two quantities related to conformational property prediction.
> 3). In Appendix J, we additionally define **Junk** rate measurement, which Intuitively  measures the fraction of generated conformations that are far away from all the conformations in the reference set.
>
> Q3: The paper attributes the advantage in RDKit to its inclusion of a force field. Therefore, the authors combine a force field with their model and assess the difference (good!) However, the conclusion in the paper seems misguided - looking at Table 1, including the force field does not result in a significant change in their model's performance. In fact, this experiment was also done in the CVGAE paper (Mansimov et al. 2019).
> A3: Firstly, on the Drugs dataset, our method improved a lot when combined with force field (FF), and achieved impressive results (COV 92.41 and MAT 0.77). Because the molecules in Drugs are much larger and challenging, actually these results are more convincing than the results on the QM9 dataset where the improvement is marginal.
> Secondly, on the QM9 dataset, though the improvement of our method is kind of limited, FF still boosts the performance of our method, which still outperforms the baselines. And the improvement of FF for other models (CVGAE and GraphDG) are still obvious.
> Thus we draw the conclusion in the paper that: 1) FF plays a vital role in generating more realistic structures, and 2) our method has a strong capacity to generate high-quality initial coordinates.

---

> ### Author Response · Authors · 2020-11-17
> **Response to the AnonReviewer1 cont.**
>
> Q4: In Table 1, the EBM improves performance on the conformation generation tasks. However, in Table 2, the EBM decreases performance on the distribution over distances task. I would have liked to see more investigation or discussion of why this is the case.
> A4:
> 1). This phenomenon is natural because typically ETM will sharpen the generated distribution towards the stable conformations with local minimal energy. By contrast, the ISO17 dataset consists of snapshots of molecular dynamics where the structures are not equilibrium conformations but samples from the density around the equilibrium state. Therefore, ETM will slightly hurt the results.
> Actually this phenomenon is also consistent with the observations for RDKit. Instead of generating unbiased samples from the underlying distribution, RDKit will also generate the stable ones with local minimal energy by involving the hand-designed molecular force field ((Simm et al.)). And as shown in the results, though highly competitive in Tab.1, RDKit also suffers much weaker results in Tab.2.
> We have refined our statement and added the above discussion in our paper.
> 2). We also follow Mansimov et al. to add a new experiment to measure the diversity of generated conformations. The results are measured by calculating the mean and standard deviation of the pairwise RMSD between each pair of generated conformations per molecule. The results are summarized as follows:
> Metric & RDKit & CVGAE & GraphDG & CGCF & CGCF+ETM
> Mean & 0.083 & 0.207 & 0.249 & 0.810 & 0.741
> Std & 0.054 & 0.187 & 0.104 & 0.223 & 0.206
> We can observe that ETM will slightly hurt the diversity of CGCF, which verifies our statement that ETM will sharpen the generated distribution towards the stable conformations with local minimal energy.
>
> Q5: The RDKit baseline is not included in Table 2. It is included in the GraphDG paper and should be included here as well.
> A5: Actually we have included RDKit as a baseline : )
>
> &nbsp; &nbsp; &nbsp; &nbsp;
>
> Reference:
> Simm et al. Simm G N C, Hernández-Lobato J M. A generative model for molecular distance geometry[J]. arXiv preprint arXiv:1909.11459, 2019.

---

### Official Review · AnonReviewer2 · 2020-10-29
**Interesting new generative model for molecular conformations with strong results; Some questions on evaluation**

**Rating:** 6
**Confidence:** 3

**Review:**

This paper presents an approach to generate diverse small molecule conformations given its graph by combining a conditional flow-based model with an energy-based model. Sampling is performed in two separate stages: 1) a normalizing flow produces a distribution over interatomic distances (which is then postprocessed into cartesian coordinates), 2) sampled coordinates are refined by Langevin dynamics with gradient signal produced from an energy-based model. The models are trained separately.

I thought that this was an interesting, principled approach that advances the state of the art in generative models for molecular conformation sampling.  This approach (CGCF+ETM) appears to generate better samples than the VAE-based baselines, though all methods are still improved by a final refinement step with traditional forcefields (esp. for larger drug-like molecules) which suggests room for improvement. The general sampling strategy is potentially relevant in many other domains.

There were some aspects of the evaluation that could be improved/clarified:
- COV and MAT do not appear to measure false positives — the CGCF+ETM approach could be generating many junk conformations and this would not be captured by the COV/MAT.  Can the authors report a statistic over the generated distribution (instead of over the test set)?
- The abstract/introduction focuses on the computational expense of competing approaches. Can the authors comment on the computationally expense for generating each conformation?
- How do the molecular properties of the generated distribution compare to the test distribution (e.g. small molecule properties in Simm & Hernandez-Lobato Table 2)?
- Can the authors clarify if any of the test set molecules from GEOM-QM9 or GEOM-DRUGS are in the training set? It was unclear from the writing.
- Which bonds are in the set of interatomic distances? The authors defines distances as the set of all covalent bonds in the molecule in the methods section, but later mention auxiliary angle/dihedral bonds. This was confusing in the first read.
- Typo in Eq (9) ‘==‘

---

> ### Author Response · Authors · 2020-11-17
> **Response to the AnonReviewer2**
>
> Thanks for your constructive comments and suggestions. The response to your concerns are listed below:
>
> Q1: COV and MAT do not appear to measure false positives — the CGCF+ETM approach could be generating many junk conformations and this would not be captured by the COV/MAT. Can the authors report a statistic over the generated distribution (instead of over the test set)?
> A1: This is a great point! We have added two experiments in the updated version to address your concern.
> 1. Since the COV and MAT score do not appear to explicitly measure the generated false samples, we additionally define **Junk** rate measurement. Intuitively, JUNK measures the fraction of generated conformations that are far away from all the conformations in the reference set. For each conformation in the generated set, it will be marked as a false sample if its RMSD to all the conformations of the reference set are above a given threshold. Typically, a lower JUNK rate means better generated quality. The results are as follows:
> Model & QM9-Mean & QM9-Median & Drugs-Mean & Drugs-Median
> CVGAE & 71.59 & 100.00 & 100.00 & 100.00
> GraphDG & 61.25 & 66.26 & 97.83 & 100.00
> CGCF & 55.24 & 57.24 & 77.82 & 90.00
> CGCF+ETM & 52.15 &  54.23 &  75.81 & 88.64
> RDKit & 17.07 & 5.90 & 45.51 & 45.94
> CVGAE + FF & 62.92 & 71.21 & 72.01 & 78.44
> GraphDG + FF & 45.53 & 46.35 & 55.50 & 61.54
> CGCF+FF & 43.01 & 46.69 & 37.48 & 36.63
> CGCF+ETM+FF & 41.63 &  43.97 & 36.16 & 33.05
> As shown in the table, our CGCF model can already outperform the existing state-of-the-art baselines with an obvious margin. The results are further improved when combined with ETM to explicitly incorporate the long-range correlations.
> 2. We follow Mansimov et al. to add a new experiment to measure the diversity of generated conformations, which is a statistic over the generated distribution instead of over the test set. The results (on ISO17) are measured by calculating the mean and standard deviation of the pairwise RMSD between each pair of generated conformations per molecule. The results are summarized as follows:
> Metric & RDKit & CVGAE & GraphDG & CGCF & CGCF+ETM
> Mean & 0.083 & 0.207 & 0.249 & 0.810 & 0.741
> Std & 0.054 & 0.187 & 0.104 & 0.223 & 0.206
> The results demonstrate that while our method can achieve the lowest MMD for distance distribution, it does not collapse to generating extremely similar conformations.
>
> Q2: The abstract/introduction focuses on the computational expense of competing approaches. Can the authors comment on the computationally expense for generating each conformation?
> A2: Actually the computational cost of Molecular Dynamics (MD) mentioned in the abstract/introduction is not comparable to deep generative models, including CVGAE, GraphDG and our method. Our deep generative models can generate a conformation in seconds, while the conventional methods typically take several hours to predict a geometry, or even days for larger ones.
>
> Q3: How do the molecular properties of the generated distribution compare to the test distribution (e.g. small molecule properties in Simm & Hernandez-Lobato Table 2)?
> A3: We follow Axelrod et al. to add an experiment on two quantities related to conformational property. In practice, we first pretrain a SchNet for energy prediction, and then take the pretrained model to estimate the energy of conformations generated from different methods. For evaluation, the first quantity is the average conformational energy <E>, which is calculated by averaging the energy of generated conformations for each molecule. And the second quantity is the lowest-energy E_L, which is defined with respect to the lowest-energy conformer for each molecule. Similar to Simm et al., due to the poor generated quality, we could not compute the properties for CVGAE, and thus this baseline is excluded from this analysis. We calculate the Mean Absolute Error (MAE) in energy properties between ground truth and generated conformations from different methods.The results are shown as follows:
> methods & E_L & <E>
> RDKit & 0.01255 & 2.60450
> GraphDG & 0.01265 & 3.13825
> CGCF & 0.01253 & 3.14391
> CGCF+ETM & 0.01250 & 3.13799
> It can be seen that different methods perform similarly. For the deep generative models, our method can achieve slightly better performance than GraphDG.
>
> Q4: Can the authors clarify if any of the test set molecules from GEOM-QM9 or GEOM-DRUGS are in the training set? It was unclear from the writing.
> A4: No, the training set and test set are totally different. We have added these details in the updated version.

---

> ### Author Response · Authors · 2020-11-17
> **Response to the AnonReviewer2 cont.**
>
> Q5: Which bonds are in the set of interatomic distances? The authors define distances as the set of all covalent bonds in the molecule in the methods section, but later mention auxiliary angle/dihedral bonds. This was confusing in the first read.
> A5: Since the bonds existing in the molecular graph are not sufficient to characterize a conformation, we pre-process the graphs by extending \textit{auxiliary} edges. Specifically, the atoms that are $2$ or $3$ hops away are connected with \textit{virtual bonds}, labeled differently from the real bonds of the original graph. These extra edges contribute to reducing the degrees of freedom in the 3D coordinates, with the edges between second neighbors helping to fix the angles between atoms, and those between third neighbors fixing dihedral angles.
> We have added contents to mention this procedure in the updated version.
>
> Reference:
> (Mansimov et al.) Mansimov E, Mahmood O, Kang S, et al. Molecular geometry prediction using a deep generative graph neural network[J]. Scientific reports, 2019, 9(1): 1-13.
> (Axelrodet al.) Axelrod S, Gomez-Bombarelli R. GEOM: Energy-annotated molecular conformations for property prediction and molecular generation[J]. arXiv preprint arXiv:2006.05531, 2020.
> (Simm et al.) Simm G N C, Hernández-Lobato J M. A generative model for molecular distance geometry[J]. arXiv preprint arXiv:1909.11459, 2019.

---

### Official Review · AnonReviewer4 · 2020-10-29
**Novel use of continuous flow in molecular conformation generation**

**Rating:** 7
**Confidence:** 4

**Review:**

The authors of this manuscript proposed a generative dynamics system for the modelling and generation of 3D conformations of molecules. Specifically, there are three components: (1) conditional graph continuous flow (CGCF) to transform random noise to distances,  (2)a closed-form distribution p(R|d, G), and (3) an energy-based tilting model (ETM) to capture long-range interactions and correct the position matrix distribution. The proposed framework was compared with two deep learning methods for conformation generations -- CVGAE & GraphDG, as well as the computational chemistry tool RDKit on GEOM-QM9, GEOM-Drugs, and ISO17 data sets. Comparisons in terms of COV and MAT scores show that the proposed method (particularly the one enhanced with ETM) can outperform baselines. Further comparisons of distances densities show that CGCF (but without ETM) worked best over baselines.

Overall, I think it is an interesting work. The major novelty is the use of continuous flow to model the conditional distribution of the distances and an energy-based model to correct the conditional distribution of positions. However, I have the following concerns.

1. The presentation of this paper can be significantly improved. A few typos need to be corrected:
section x -? Section X
a optimization -> an optimization
which a ...-> which is a...
demotes -> denotes
references should be further formatted

2. A bit more precise description about the force-fields algorithms in RDKit is needed.

3. From results in Table 2, CGCF combined with the ETM component does work better than GraphDG, although the authors state that it is because the sharpness of the distance distribution. Clear justifications should be given to show the benefits, if any, of this phenomenon.

---

> ### Author Response · Authors · 2020-11-17
> **Response to the AnonReviewer4**
>
> Thanks for your constructive comments and suggestions. The response to your concerns are listed below:
>
> Q1: The presentation of this paper can be significantly improved. A few typos need to be corrected: section x -? Section X a optimization -> an optimization which a ...-> which is a... demotes -> denotes references should be further formatted
> A1: Thanks for pointing them out and we’ve already corrected them in the updated version.
>
> Q2: A bit more precise description about the force-fields algorithms in RDKit is needed.
> A2: In MMFF, the energy expression is constituted by seven terms: bond stretching, angle bending, stretch-bend, out-of-plane bending, torsional, van der Waals and electrostatic. The detailed functional form of individual terms can be found in the original literature. To build the force field for a given molecular system, the first step is to assign correct types to each atom. At the second step, atom-centered partial charges are computed according to the MMFF charge model. Then, all bonded and non-bonded interactions in the molecular system under study, depending on its structure and connectivity, are loaded into the energy expression. Optionally, external restraining terms can be added to the MMFF energy expression, with the purpose of constraining selected internal coordinates during geometry optimizations. Once all bonded and non-bonded interactions, plus optional restraints, have been loaded into the MMFF energy expression, potential gradients of the system under study can be computed to minimize the energy.
> We have added the above detailed description of the MMFF force field algorithm in the updated version.
>
> Q3: From results in Table 2, CGCF combined with the ETM component does work better than GraphDG, although the authors state that it is because of the sharpness of the distance distribution. Clear justifications should be given to show the benefits, if any, of this phenomenon.
> A3:
> 1. Tab.2 shows that generally CFCF+ETM works better than GraphDG, but worse than CGCF itself. Here we give a discussion about why ETM will slightly hurt the performance and its benefits:
> This phenomenon is natural because typically ETM will sharpen the generated distribution towards the stable conformations with local minimal energy. By contrast, the ISO17 dataset consists of snapshots of molecular dynamics where the structures are not equilibrium conformations but samples from the density around the equilibrium state. Therefore, ETM will slightly hurt the results.
> Actually this phenomenon is also consistent with the observations for RDKit. Instead of generating unbiased samples from the underlying distribution, RDKit will also generate the stable ones with local minimal energy by involving the hand-designed molecular force field (Simm et al.). And as shown in the results, though highly competitive in Tab.1, RDKit also suffers much weaker results in Tab.2. Therefore, similarly to RDKit, the benefits of the sharpness of the generated distribution can be found in Tab.1, where ETM can obviously boost the performance of CGCF by generating more stable conformations.
> We have refined our statement and added the above discussion in the updated version.
> 2. We also follow Mansimov et al. to add a new experiment to measure the diversity of generated conformations. The results are measured by calculating the mean and standard deviation of the pairwise RMSD between each pair of generated conformations per molecule. The results are summarized as follows:
> Metric & RDKit & CVGAE & GraphDG & CGCF & CGCF+ETM
> Mean & 0.083 & 0.207 & 0.249 & 0.810 & 0.741
> Std & 0.054 & 0.187 & 0.104 & 0.223 & 0.206
> We can observe that ETM will slightly hurt the diversity of CGCF, which verifies our statement that ETM will sharpen the generated distribution towards the stable conformations with local minimal energy.
>
> &nbsp; &nbsp; &nbsp; &nbsp;
>
> Reference:
> (Simm et al.) Simm G N C, Hernández-Lobato J M. A generative model for molecular distance geometry[J]. arXiv preprint arXiv:1909.11459, 2019.
> (Mansimov et al.) Mansimov E, Mahmood O, Kang S, et al. Molecular geometry prediction using a deep generative graph neural network[J]. Scientific reports, 2019, 9(1): 1-13.

---

### Public Comment · ~Jianwen_Xie1 · 2020-11-14
**related EBM works before 2019**

Dear Authors and Reviewers,

We found that the current paper missed some important references about pioneering works that are related to energy-based generative models parameterized with deep net energy.

The first paper that proposes to train an energy-based model parameterized by modern deep neural network and learned it by Langevin based MLE is in (Xie. ICML 2016) [1]. The model is called generative ConvNet, because it can be derived from the discriminative ConvNet. This is also the first paper to formulate modern ConvNet-parametrized EBM as exponential tilting of a reference distribution, and connect it to discriminative ConvNet classifier. That is, EBM is a generative version of a discriminator. (Xie. ICML 2016) [1] originally studied such an EBM model on image generation theoretically and practically in 2016.

(Xie. CVPR 2017) [2] (Xie. PAMI 2019) [3] proposed to use Spatial-Temporal ConvNet as the energy function in EBMs for video generation. The model is called Spatial-Temporal generative ConvNet.

(Xie. CVPR 2018) [4] also proposed to use volumetric 3D ConvNet as the energy function for 3D shape pattern generation. It is called 3D descriptor Net.

Also, the Generative Cooperative Nets (CoopNets) (Xie. PAMI 2018)[5] and (Xie. AAAI 2018) [6], which jointly trains an EBM and a generator network by MCMC teaching.

Those are the more original and earlier papers for deep EBMs with ConvNet as energy function than what you have cited, e.g., [7](Yilun Du and Igor Mordatch, 2019).

References:

[1] A Theory of Generative ConvNet. Jianwen Xie *, Yang Lu *, Song-Chun Zhu, Ying Nian Wu (ICML 2016)

[2] Synthesizing Dynamic Pattern by Spatial-Temporal Generative ConvNet Jianwen Xie, Song-Chun Zhu, Ying Nian Wu (CVPR 2017)

[3] Learning Energy-based Spatial-Temporal Generative ConvNet for Dynamic Patterns Jianwen Xie, Song-Chun Zhu, Ying Nian Wu IEEE Transactions on Pattern Analysis and Machine Intelligence (TPAMI) 2019

[4] Learning Descriptor Networks for 3D Shape Synthesis and Analysis Jianwen Xie *, Zilong Zheng *, Ruiqi Gao, Wenguan Wang, Song-Chun Zhu, Ying Nian Wu (CVPR) 2018

[5] Cooperative Training of Descriptor and Generator Networks. Jianwen Xie, Yang Lu, Ruiqi Gao, Song-Chun Zhu, Ying Nian Wu. IEEE Transactions on Pattern Analysis and Machine Intelligence (TPAMI) 2018

[6] Cooperative Learning of Energy-Based Model and Latent Variable Model via MCMC Teaching. Jianwen Xie, Yang Lu, Ruiqi Gao, Ying Nian Wu. AAAI 2018.

[7] Yilun Du and Igor Mordatch. Implicit generation and modeling with energy based models. In Advances in Neural Information Processing Systems, pages 3603–3613, 2019

Thank you!

---

> ### Author Response · Authors · 2020-11-17
> **Thank you very much for pointing out the related work!**
>
> Thank you very much for pointing out the related work! Our work is indeed related to these works, and we have added the suggested references in the updated version. However, our work is fundamentally different from these works.
>
> Different from the previous works, we concentrate on molecular geometry generation, and propose a novel and principled probabilistic framework to address the domain-specific problems. More specifically, we first predict the atomic distances through the continuous normalizing flow, and then convert them to the desired 3D conformation and optimize it with the energy-based model. This procedure enables us to keep the rotational and translational invariance property. Besides, to the best of our knowledge, we are the first one to combine neural ODE with EBMs. We take the ODE model to improve the training of EBM, and combine both to conduct the two-stage sampling dynamics.

---

### Author Response · Authors · 2020-11-17
**General response to all the reviewers and readers**

We would like first to thank all the reviewers and readers for your constructive reviews.
We’ve revised the paper according to your reviews. Specifically, we have made the following changes:

1). We have added three experiments:
1. In Section 4.3, we follow Mansimov et al. to add a new experiment to measure the diversity of generated conformations, which is a statistic over the generated distribution instead of over the test set. The results (on ISO17) are measured by calculating the mean and standard deviation of the pairwise RMSD between each pair of generated conformations per molecule.
2. In Appendix J, we follow Axelrod et al. to add an experiment on two quantities related to conformational property prediction.
3. In Appendix J, we additionally test **Junk** rate measurement, which intuitively measures the fraction of generated conformations that are far away from all the conformations in the reference set.

2). In Section 4.3, we add discussions and experiments on why EBM slightly decreases performance on the distribution over distances task.

3). In Appendix A, we add the suggested references and discuss the difference between our work and previous related works for computer vision.

4). We add more description about the data processing procedure, and more information about the experiment details (the training and testing data setting).

&nbsp;&nbsp;&nbsp;&nbsp;

Reference:
(Mansimov et al.) Mansimov E, Mahmood O, Kang S, et al. Molecular geometry prediction using a deep generative graph neural network[J]. Scientific reports, 2019, 9(1): 1-13.
(Axelrodet al.) Axelrod S, Gomez-Bombarelli R. GEOM: Energy-annotated molecular conformations for property prediction and molecular generation[J]. arXiv preprint arXiv:2006.05531, 2020.

---

### Decision · Program_Chairs · 2021-01-07
**Final Decision**

**Decision:**

Accept (Poster)

**Comment:**

The paper combines flow-based and energy-based models to generate molecular conformations given a molecular graph.
For this, a continuous flow model is used to map the graph-based molecular representation into a distribution over conformations.
An energy-based model (EBM) is used to further help the model capture long-range atomic interactions. The proposed method is compared with strong baselines: CVGAE, GraphDG, and RDKit.

The authors addressed most of the reviewers' concerns in the rebuttal.

All the reviewers agree on acceptance.